# New Insights into the Pathogenesis of Giant Cell Arteritis: Mechanisms Involved in Maintaining Vascular Inflammation

**DOI:** 10.3390/jcm11102905

**Published:** 2022-05-20

**Authors:** Hélène Greigert, Coraline Genet, André Ramon, Bernard Bonnotte, Maxime Samson

**Affiliations:** 1Department of Internal Medicine and Clinical Immunology, Dijon University Hospital, 14 Bd. Gaffarel, 21000 Dijon, France; helene.greigert@chu-dijon.fr (H.G.); bernard.bonnotte@chu-dijon.fr (B.B.); 2Department of Vascular Medicine, Dijon University Hospital, 14 Bd. Gaffarel, 21000 Dijon, France; 3INSERM, EFS BFC, UMR1098, RIGHT Interactions Greffon-Hôte-Tumeur/Ingénierie Cellulaire et Génique, Université Bourgogne Franche-Comté, 21000 Dijon, France; coraline.genet@u-bopurgogne.fr (C.G.); andre.ramon@chu-dijon.fr (A.R.); 4Department of Rheumatology, Dijon University Hospital, 14 Bd. Gaffarel, 21000 Dijon, France

**Keywords:** giant cell arteritis, pathogenesis, IL-6, T cells, GM-CSF, vascular smooth muscle cells

## Abstract

The giant cell arteritis (GCA) pathophysiology is complex and multifactorial, involving a predisposing genetic background, the role of immune aging and the activation of vascular dendritic cells by an unknown trigger. Once activated, dendritic cells recruit CD4 T cells and induce their activation, proliferation and polarization into Th1 and Th17, which produce interferon-gamma (IFN-γ) and interleukin-17 (IL-17), respectively. IFN-γ triggers the production of chemokines by vascular smooth muscle cells, which leads to the recruitment of additional CD4 and CD8 T cells and also monocytes that differentiate into macrophages. Recent data have shown that IL-17, IFN-γ and GM-CSF induce the differentiation of macrophage subpopulations, which play a role in the destruction of the arterial wall, in neoangiogenesis or intimal hyperplasia. Under the influence of different mediators, mainly endothelin-1 and PDGF, vascular smooth muscle cells migrate to the intima, proliferate and change their phenotype to become myofibroblasts that further proliferate and produce extracellular matrix proteins, increasing the vascular stenosis. In addition, several defects in the immune regulatory mechanisms probably contribute to chronic vascular inflammation in GCA: a defect in the PD-1/PD-L1 pathway, a quantitative and qualitative Treg deficiency, the implication of resident cells, the role of GM-CSF and IL-6, the implication of the NOTCH pathway and the role of mucosal-associated invariant T cells and tissue-resident memory T cells.

## 1. Introduction

Giant cell arteritis (GCA) is a granulomatous vasculitis affecting large vessels, especially the aorta and its major branches, mainly subclavian arteries and cranial arteries of the external carotid system. This vasculitis affects adults after 50 years (peak incidence between 70 and 80 years) and women are affected two to three times more frequently than men [1,2,3,4,5]. Systemic signs, i.e., fever, asthenia, anorexia and weight loss, are frequent, unspecific and related to systemic inflammation that is largely correlated with serum interleukin-6 (IL-6) elevation [6,7]. Patients also present ischemic signs, which are responsible for the most serious features of this vasculitis and the clinical manifestations of which directly depend on the topography of the arterial involvement. Ischemic signs are a consequence of the remodeling process affecting the arterial wall, which leads to its thickening, and finally, to vascular stenosis or occlusion [8]. The diagnosis of GCA is usually confirmed by a temporal artery biopsy (TAB) revealing non-necrotizing granulomatous panarteritis and showing these two pathological processes: 1—vascular inflammation related to a transmural cellular infiltrate composed of mononuclear cells (T cells and macrophages) and sometimes giant cells, 2—vascular remodeling with typical fragmentation of the internal elastic lamina, destruction of the media and proliferation of myofibroblasts, leading to a hyperplastic neointima and resulting in the stenosis of the vascular lumen [9,10] (Figure 1).

## 2. Implications of Genetic Background, Epigenetics and Aging

The incidence of GCA follows a north to south gradient. The highest prevalence is observed in Scandinavian countries and in Olmsted County, Minnesota (overall annual incidence reaches 18.8 per 100,000 persons aged 50 years or older), where the population has a similar ethnicity, suggesting that the occurrence of GCA is favored by one genetic background [11,12,13,14].

In the last 20 years, several studies have reported genetic polymorphisms affecting genes involved in the immune response and inflammation that are associated with an increased risk of having GCA [15]. The most relevant association is with the major histocompatibility complex (MHC) class-II genes. There is a strong association between GCA and the human leukocyte antigen (HLA) region, which highlights the essential role of adaptive immunity in the pathogenesis of this vasculitis. Alleles of *HLA-DRB1*04*, particularly the *HLA-DRB1*0401*, *DRB1*0404* or *DRB1*0408* haplotypes, have been shown to be associated with the occurrence of GCA in independent cohorts [16,17,18,19,20,21,22,23,24,25,26,27] and are expressed by 60% of patients affected by polymyalgia rheumatica (PMR) or GCA [18,24,28]. These haplotypes are thought to be responsible for the selection of peptides involved in the GCA pathogenesis that are then presented to CD4^+^ T cells. Importantly, a genome-wide association study (GWAS) assessing 2134 GCA patients from 10 independent populations of European ancestry and 9125 controls confirmed that class-II MHC is the genomic region with the strongest association with GCA [29]. More recently, a study including 184 patients with cranial GCA, 105 patients with large-vessel GCA and 486 healthy controls showed an association between the *HLA-B*15:01* allele and GCA, regardless of the clinical phenotype of the disease, and that there was an increased risk of developing GCA, whatever its phenotype, if *HLA-B*15:01* and *HLA-DRB1*04:01* were expressed together [30]. This latest study reinforces the concept that susceptibility to GCA is strongly related to the HLA region. Furthermore, this GWAS study also identified two genes outside the HLA region associated with GCA susceptibility: *PLG* (gene of plasminogen) and *P4HA2* (encodes an isoform of the alpha subunit of the collagen prolyl 4-hydroxylase, which is essential for collagen biosynthesis). Interestingly, these genes are involved in a wide spectrum of physiological processes, some of which are relevant to the GCA pathogenesis, such as wound healing, angiogenesis, lymphocyte recruitment and inflammation for *PLG* and vascular remodeling for *P4HA2* [29,31].

In addition to genetic factors, research has shown that a dynamic relationship exists between genetic predispositions and environmental factors via epigenetics, which is defined as the changes in gene expression that occur without altering the underlying DNA sequence but through DNA methylation, histone modifications and microRNAs [32]. Differences in the DNA methylation level have been reported in temporal arteries of GCA patients when compared to non-GCA patients for several genes. Indeed hypomethylation, leading to an increase in the expression of genes, was identified in several pro-inflammatory genes (*IFNG*, *IL21*, *IL23R*, *IL17RA, TNF*, *IL6*, *IL1B*, *IL2*, *IL18*, *LTA*, *LTB*, *CCR7*, *CD6*, *NLRP1*), as well as *RUNX3*, which is involved in the production of IFN-γ and *CD40LG,* which encodes for CD40L, a protein involved in the cross-talk between T and B cells [33,34]. The hypomethylation of these genes correlated with the activation of T lymphocytes and their polarization into Th1 and Th17 cells, as previously described in GCA [35,36,37]. Along this line, it was demonstrated in the same study [34] that genes encoding for components of the TCR, for co-stimulatory molecules (*CD28*) and proteins implicated in T-cell activation pathways (*NFATC1*, *NFATC2*, *PPP3CC*), were also hypomethylated in GCA arteries when compared to non-GCA arteries [34].

Age is an essential factor in the onset of GCA, which occurs after 50 years with an increasing incidence. This could, therefore, naturally be a major etiological factor in the GCA pathogenesis [38]. The aging process is intimately related to epigenetics. The level of DNA methyltransferase 1 (DNMT1), which is the enzyme implicated in the maintenance of DNA methylation at each cell division, declines with age in T cells, causing aberrant methylation profiles that can lead to malignant transformation or autoimmunity, such as with myelodysplastic syndromes or VEXAS, which is now the archetype of the acquired auto-inflammatory disease occurring in elderly men [32,39,40,41].

Furthermore, the aging process has been associated with modifications to multiple cells implicated in the immune response and vascular remodeling, such as dendritic cells (DC), T cells, endothelial cells and vascular smooth muscle cells (VSMC) [38]. In particular, aging triggers a decrease in the number of naive T cells, increase in memory and effector T cells, decrease in the diversity of the T-cell repertoire and enrichment in CD4^+^CD28^−^ and CD8^+^CD28^−^ senescent T cells [42,43,44,45]. Immune aging also alters the regulation of immune cells, for instance, by impacting CD8^+^ Treg [46], thus leading to the production of pro-inflammatory cytokines (IL-1β, IL-6 and TNF-α) by senescent DCs, macrophages, endothelial cells and fibroblasts [38,47]. During aging, DCs still express toll-like receptor (TLR), but their activation and migration abilities are impaired [48,49]. This aging process might generate chronic inflammation, leading to the occurrence of atherosclerosis and also autoimmune diseases such as GCA. The aging process also modifies arterial tissue, the target of GCA, by medial degeneration, calcium deposition, increased stiffness, wall thickening, elastic fiber fractures and biochemical modifications of matrix proteins [50,51,52,53]. Combined with this proinflammatory state, these modifications could trigger immunization against arterial auto-antigens and lead to the occurrence of vasculitis [38]. Similarly, a recent study showed that aging and long-term alterations to the immune system are associated with susceptibility to developing GCA, which is also suggested by the association between a history of infections and an increased incidence of GCA [54]. Thus, aging of the immune system could explain, among other things, the association between GCA and varicella-zoster virus (VZV) infections without this virus playing a real causal role.

## 3. Potential Role of Infections in the Initiation of GCA

Several studies have investigated the potential roles of a large number of viruses and/or bacteria in the pathogenesis of GCA. The condition appears to have a seasonal variation, suggesting the involvement of an infectious trigger in the induction of this vasculitis [4,55]. Many case-controlled studies have compared the level of viral or bacterial DNA in TAB (PCR, immunohistochemistry or in situ hybridization) between patients with biopsy-proven GCA and controls. These studies found an association between GCA and the presence of cytomegalovirus, parvovirus B19, herpes simplex virus, human parainfluenza 1 and *Chlamydia pneumonia,* but neither Epstein-Barr virus nor human herpesvirus 6 [56]. However, the results of these studies often contradicted others and they have never been confirmed in large cohorts [56]. Recently, VZV has been suggested as a triggering factor for GCA [57]. VZV is an exclusively human neurotropic virus, which causes chickenpox and zoster and is also able to replicate in arteries, especially cerebral arteries, thus being able to induce vasculitis [58]. This vasculitis, the histological appearance of which is very similar to GCA, affects large and medium arteries and can lead to stenosis, occlusion, thrombosis or dissection [58,59]. VZV vasculitis can also affect the temporal arteries and cause GCA-like symptoms [60]. In contrast to earlier research [56,61,62,63,64,65,66], a recent study suggested that VZV may be the triggering agent of GCA because the presence of VZV was detected in 73% of positive and 64% of negative TAB from patients with GCA, compared to only 22% of negative TAB from controls [67]. Nevertheless, the implication of VZV in the pathogenesis of GCA remains unclear since these data were not confirmed in a more recent study [68]. Thus, the occurrence of zoster is infrequent in GCA [69], even when immunosuppressive therapy is used [70,71,72,73,74,75,76,77,78,79]. Plus, the antibody that was used for VZV staining for immunohistochemistry seemed to lack specificity because it cross-reacted with antigens expressed by VSMC [80]. Rather than a cause of GCA, the detection of VZV in TAB from GCA patients could just be related to immune aging that decreases the control of the replication of this virus.

The role of DC in the pathogenesis of GCA suggests a relationship between a potential infection and the initiation of GCA. It is generally assumed that an infectious agent can activate adventitial DC and trigger immunologic processes leading to the development of vasculitis. However, this hypothesis has never been fully resolved, mainly because no specific pathogen triggering GCA has been identified. Indeed, very recent work analyzing the microbiome of temporal arteries or thoracic aortic aneurysms due to GCA provided contradictory and inconclusive results [81,82]. It is quite possible that the precise nature of this potential infectious agent, providing a danger signal for DC, does not matter but that, instead, the essential point is that this activation occurs in a predisposed genetic background favoring the implementation of mechanisms leading to the occurrence of vasculitis and its persistence due to defects in the regulation of the immune system. This is a notion to which we will return in this review.

## 4. Immunopathological Model of GCA

The immunopathological model of GCA can be divided into four main phases (Figure 2).

### 4.1. Phase 1: Loss of Tolerance and Activation of Resident Dendritic Cells of the Adventitia

Immature myeloid DCs (S100^+^CD11c^+^CCR6^+^CD83^−^MHC-II^low^) are physiologically localized in the adventitia where they are involved in immune surveillance [8,83,84]. These cells can trigger adaptive immunity after detecting danger signals through pattern recognition receptors (PPRs) such as TLR. By contrast, they induce tolerance in the absence of a danger signal [85]. The detection of a danger signal via TLR of adventitial DCs induces their activation, followed by phenotypic modifications (S100^+^CD11c^+^CCR7^+^CD83^+^CD80/86^+^MHC-II^high^) and the production of cytokines and chemokines responsible for the recruitment of CD4^+^ T cells in the arterial wall. Once they are activated, DCs express high levels of class-II major histocompatibility complex (MHC-II) and co-stimulatory molecules (CD80 and CD86), making them capable of activating CD4^+^ T cells. In GCA, DCs produce CCL19 and CCL21 chemokines and express their receptor (CCR7) so that they are trapped in the arterial wall instead of migrating into satellite lymph nodes, as is the case in classical immune responses [83].

Adventitial DCs play an essential role in initiating the GCA pathogenesis. It has been demonstrated in a model of immunocompromised mice engrafted with temporal arteries that DC activation, mainly by TLR4 ligands, is needed to trigger the recruitment of T cells in the arterial wall. A major role of DCs in the GCA pathogenesis was also shown by how the depletion of CD83 cells (i.e., activated DCs and macrophages) in this model resulted in a significant decrease in lesions of vasculitis and IFN-γ mRNA expression [84]. The tropism of GCA for cranial arteries could also be related to characteristics of arterial DCs as it was demonstrated that TLR expression varied across arterial territories and that the TLR expression profiles of arteries typically affected by GCA were very similar [86].

### 4.2. Phase 2: Recruitment, Activation and Polarization of CD4^+^ T Cells

CD4^+^ T cells, which are physiologically absent from the walls of healthy arteries, play a major role in the pathogenesis of GCA, as highlighted by the fact that their depletion in immunocompromised mice engrafted with GCA arteries strongly decreases vasculitis lesions [87]. Studies have also shown a restricted oligoclonal repertoire of T-cell receptors (TCRs) in the arterial wall, thus indicating an antigen-mediated proliferation of CD4^+^ T cells [88,89,90]. The activation of T cells in GCA is also highlighted by NFAT expression and its localization to the nuclei of cells in GCA lesions [34]. In GCA, CD4^+^ T cells are recruited through the chemokines (CCL18, CCL19, CCL20 and CCL21) produced by activated DCs [8,56]. Among these chemokines, CCL20 plays a major role as it triggers the recruitment of CD4^+^ T cells expressing CCR6, the ligand of CCL20. Interestingly, the nature of the initial danger signal determines the architecture of the vasculitis. TLR4 ligands trigger the recruitment of CD4^+^CCR6^+^ T lymphocytes, a phenotype matching Th17 cells [36,91,92,93], and then infiltrate all the layers of the artery, leading to panarteritis, which is characteristic of GCA. By contrast, TLR5 ligands only trigger adventitial vasculitis close to *Vasa vasorum* [94].

T cells are thought to be recruited via *Vasa vasorum* of the adventitia, in which endothelial cells express adhesion molecules such as intercellular adhesion molecule-1 (ICAM-1) and vascular cell adhesion molecule-1 (VCAM-1) [8,56]. Endothelial cells are a natural barrier between blood and tissues and are involved in the regulation of vasomotion, hemostasis, angiogenesis and inflammation. In GCA, endothelial cells are activated by the cytokines produced by mononuclear cells and thus express high levels of adhesion molecules such as ICAM-1, ICAM-2, P-selectin, E-selectin and VCAM-1, which participate in the recruitment of additional immune cells [95]. Along this line, the concentration of soluble ICAM-1 in GCA patients correlates with disease activity [96].

Once recruited in the arterial wall, CD4^+^ T cells are activated by DCs that present still unidentified antigen(s). The presence of proinflammatory cytokines (IL-12, IL-18, IL-23, IL-6 and IL-1β) in the microenvironment polarizes CD4^+^ T lymphocytes toward Th1 and Th17 cells [36,37,97]. Th1 cells are generated in the presence of IL-12 and IL-18 and produce IFN-γ, whereas Th17 cells are generated in the presence of IL-6, IL-1β and IL-23 and produce IL-17 [93]. This polarization toward Th1 and Th17 rather than Th2 or Treg cells is favored by preferential recruitment of CD4^+^CD161^+^ T cells, which express CCR6 and are prone to polarize into Th17 and Th1 cells, with a high level of plasticity between these two subsets that is highlighted by the presence of a high number of double-positive cells (IFN-γ^+^IL-17^+^) [36].

IL-17 production by Th17 is rapidly decreased by corticosteroid treatment [35,36,37,98], whereas IFN-γ production by Th1 cells tends to persist, especially at lower corticosteroid doses [35], which may contribute to the persistence of smoldering vascular inflammation, explaining the occurrence of relapse in about one in two patients when corticosteroids are tapered [99].

Therefore, therapeutic strategies targeting Th1 cells more specifically seem to be required to improve the treatment of GCA. This could be achieved by therapeutic strategies inhibiting intracellular cytokine signaling such as Janus kinase (JAK) inhibitors. Environment-cell communications rely on cytokine signals that trigger the JAK and signal transducer and activator of transcription (STAT) pathway. Th1 lineage commitment is strictly linked to STAT1- and STAT4-mediated gene induction, and STAT3 is considered the master regulator for Th17 cell differentiation. The effect of tofacitinib, a JAK3 and JAK1 inhibitor, has been studied in immunocompromised mice engrafted with temporal arteries and reconstituted with T cells and monocytes from GCA patients. The authors showed that, compared to healthy arteries, GCA arteries were characterized by an increased expression of STAT1- and STAT2-targeted genes, thus highlighting the activation of the IFN-γ pathway. Furthermore, pharmacologic inhibition of JAK3 and JAK1, the latter being involved in the IFN-γ pathway, effectively decreased vascular inflammation, the Th1 immune response, neoangiogenesis and intimal hyperplasia [100].

The implication of Th17 cells, the proportion of which is increased in the blood of GCA patients and that infiltrate GCA arteries [35,36,37], contrasts with a quantitative defect of circulating Treg (CD4^+^CD25^high^FoxP3^+^) and the low expression of FoxP3 in the arteries of GCA patients [36,101]. These data support the concept of a Th17/Treg imbalance in GCA [92] and are consistent with the implication of IL-6 in the pathogenesis of this vasculitis. IL-6, which is produced in GCA lesions [98] and the concentration of which in the serum correlates with disease activity [7,36], physiologically controls the Th17/Treg balance since IL-6 and transforming growth factor-beta (TGF-β) trigger Th17 polarization, whereas TGF-β alone leads to the generation of Treg [102].

In addition to IL-6, IL-21, a pro-inflammatory cytokine mainly produced by follicular helper T cells (TFH) [103], is also produced in GCA lesions. The proportion of CD4^+^IL-21^+^ T cells is increased in the blood of GCA patients and correlates with the levels of Th1 and Th17 cells. Moreover, IL-21 increases Th1 and Th17 polarization, whereas it decreases Treg differentiation in vitro [37].

Interestingly, the predominant cytokine profile could play a role in the emergence of the various histological patterns of GCA [104,105]. It was reported that IL-17 overexpression was observed mainly in arteries with transmural inflammation and vasa vasorum vasculitis, whereas IL-9 and Th9 polarization was predominant in cases of transmural inflammation and small-vessel vasculitis [106].

### 4.3. Phase 3: Recruitment of CD8^+^ T Cells and Monocytes

The strong infiltration of Th1 and Th17 cells into the arterial wall is responsible for the production of large amounts of IFN-γ and IL-17, respectively. IFN-γ induces the production of several chemokines (CCL2, CXCL9, CXCL10 and CXCL11) by VSMCs [107]. CXCL9, CXCL10 and CXCL11 trigger the recruitment of additional Th1 and also CD8^+^ T cells expressing CXCR3 [108]. As was demonstrated for CD4^+^ T cells, we showed that the repertoire of circulating CD8^+^ T cells is also oligoclonal, thus supporting their antigen-driven activation. Furthermore, CD8^+^ T cells infiltrate the arterial wall and produce cytokines (IL-17 and IFN-γ) and cytotoxic molecules (granzymes and perforin) [108]. Through the production of cytotoxic molecules and IFN-γ, CD8^+^ T cells could be of particular importance for initiating vascular remodeling pathways, as supported by the fact that strong CD8^+^ T cell infiltration into temporal arteries is associated with more severe disease [108].

Monocytes are precursors of tissue macrophages. There are different subsets of monocytes, which differ in their phenotype, function and chemotaxis pathway. Recruitment of classical monocytes (CD14^bright^CD16^neg^) mainly depends on the CCR2-CCL2 axis, while that of non-classical monocytes (CD14^dim^CD16^+^) depends on the CX3CR1-CX3CL1 axis [109]. In the blood of GCA patients, the dynamics and distribution of monocyte subsets are altered [110,111]. In GCA lesions, IFN-γ induces the production of CCL2 by VSMC, which leads to the recruitment of monocytes that express its receptor (CCR2) and then merge to form multinucleated giant cells, the hallmark of GCA [107]. Along this line, macrophages with a phenotype resembling the classical monocyte subset and expressing CCR2 were detected in the vessel wall of GCA patients [109,111]. However, other studies have qualified this hypothesis by showing that the majority of macrophages infiltrating temporal arteries of GCA patients resemble non-classical monocytes with CD16 and CX3CR1 expression, but often lack CCR2 expression [111,112], suggesting that both classical and non-classical monocytes are involved in GCA and that further studies focusing on monocytes and macrophages are needed.

### 4.4. Phase 4: Vascular Remodeling

Ischemic signs of GCA are related to the progressive narrowing of vascular lumen in affected arteries, which is the consequence of the remodeling process involving the arterial wall. This process is characterized by the destruction of the media and the emergence of a neo-intima made of myofibroblasts and extracellular matrix proteins, resulting in intimal hyperplasia and vessel occlusion (Figure 1) [4].

Macrophages play a major role in vascular remodeling through their ability to produce enzymes, growth factors and other mediators. Two types of macrophages are usually distinguished. M1 macrophages, which express CD64, are induced by IFN-γ and exert pro-inflammatory functions thanks to the production of proinflammatory cytokines, growth factors, matrix metalloproteinases (MMP) and reactive oxygen species (ROS). In contrast, M2 macrophages, which express CD206 and FRβ, have anti-inflammatory and tissue-repairing functions. Like T cells, macrophages are highly plastic, and their heterogeneous phenotype has been reported in GCA [112,113]. It is important to note that the distinction between M1 and M2 macrophages is based on in vitro data and that under pathological conditions, this distinction is not as clear-cut. In addition, cytokines produced by T cells, such as IFN-γ, IL-17 or GM-CSF, can affect the phenotypes of macrophages, with more complex phenotypes appearing than just M1 and M2 [112]. IFN-γ primes pro-inflammatory macrophages that produce IL-1β, IL-6, TGF-β1 and PDGF. IL-17 induces CD163 expression, which is indicative of M2-type polarization. GM-CSF can skew macrophages toward CD206^+^ ones that produce YKL-40, MMP-9 and M-CSF [112].

Macrophages play a critical role in vascular remodeling by promoting angiogenesis, intimal hyperplasia and tissue destruction. A recent study investigated the special distribution of different macrophage phenotypes in GCA arteries and showed that CD206^+^MMP-9^+^ macrophages are located at the site of tissue destruction, whereas FRβ^+^ macrophages are located in the inner intima of arteries with degrees of intimal hyperplasia, and that this pattern was specific to GCA lesions and not seen in atherosclerotic lesions. Further experiments showed that GM-CSF upregulated CD206 expression, whereas FRβ was increased in M-CSF-skewed macrophages [114]. This led the authors to hypothesize that, once recruited in the arterial wall, monocytes are exposed to GM-CSF produced by T and B lymphocytes and endothelial cells, and thus differentiate into CD206^+^ macrophages that produce MMP9 involved in the destruction of the arterial wall. These CD206^+^ macrophages can also produce M-CSF, which induces the expression of FRβ that produce mediators, mainly PDGF-AA, promoting myofibroblast migration and proliferation and thus intimal hyperplasia, leading to ischemic signs of GCA [114].

In addition, the same group demonstrated an increase in YKL-40 produced by the CD206^+^MMP9^+^ macrophage subset in GCA inflamed temporal arteries [115]. Consistent with their previous results, they showed in vitro that GM-CSF increased the production of YKL-40 more than M-CSF in macrophages from GCA patients, but not from healthy controls. Functional experiments showed that knockdown of YKL-40 with siRNA led to a significant reduction in MMP-9 production by macrophages. In contrast, YKL-40 increased neoangiogenesis as assessed by tube formation by HUVECS. Taken together, these results suggest that targeting GM-CSF, YKL-40 or its receptor (IL-13Rα2) is promising to control vascular remodeling in GCA [115].

MMPs are also key players in vascular remodeling. MMP-2 and MMP-9 produced by VSMCs and macrophages degrade elastin and destroy cellular matrix proteins, causing the destruction of the media and digestion of the internal elastic lamina [56,116]. MMP-2 and MMP-9 are the main MMP detected in GCA lesions, essentially in macrophages and giant cells adjacent to the internal elastic lamina [113,117,118]. As mentioned before, recent studies indicate that MMP-9 is mainly produced by CD206^+^YKL-40^+^MMP-9^+^ macrophages localized in the media and media borders where elastic lamina degradation takes place, and that these cells support tissue destruction and neoangiogenesis [112,115]. MMP-9 also plays a major role in large-vessel vasculitis by controlling the access of monocytes and T cells to the vascular wall since T cells depend on MMP-9 producing monocytes to pass through collagen IV membranes. This major role of MMP-9 in GCA was confirmed using a human artery-SCID-mouse model in which wall inflammation was induced in human vessels engrafted to immunodeficient mice that were immunoreconstituted with PBMC from patients with GCA. In this model, mice receiving anti-MMP-9 treatment for two weeks had a dramatic decrease in all vasculitis processes (T-cell and monocyte infiltration, neoangiogenesis, neointimal growth) [119].

Current knowledge also suggests that IFN-γ, which is produced by Th1 cells, is the main lymphocytic cytokine inducing vascular remodeling. IFN-γ-activated macrophages, giant cells or injured VSMCs produce growth factors, essentially platelet-derived growth factor (PDGF) and vascular endothelial growth factor (VEGF) [120]. PDGF is implicated in the activation and proliferation of VSMCs and their migration toward the intima, thus resulting in intimal hyperplasia. In GCA, VSMCs are injured by mediators released by mononuclear cells, which have accumulated in the media, and acquire pro-inflammatory properties [121]. Activated macrophages and VSMCs themselves produce several growth factors (PDGF, TGF-β, endothelin-1 [ET-1], NGF and BDNF neutrophins [98,121,122,123]), thus inducing the migration of VSMCs into the intima and their differentiation into myofibroblasts, which synthesize matrix proteins. This process finally leads to intimal hyperplasia and vascular occlusion. VSMCs also produce MMP-9 and especially MMP-2, which allow them to destroy the media and the internal elastic lamina [94,124], thus facilitating their migration into tissues. Furthermore, the blockade of the PDGF receptor with imatinib results in a significant decrease in the proliferation of VSMCs from ex vivo cultured temporal arteries [121]. ET-1 has also been shown to be implicated in vascular remodeling during GCA. In normal conditions, ET-1 is produced by endothelial cells and VSMCs. In GCA, the ET-1 pathway is upregulated since ET-1 is expressed by leukocytes and VSMCs and as there is upregulation of the expression of receptors A and B of ET-1 by VSMCs [125,126]. The blockade of ET-1 receptors (A and/or B) decreases the migration [126] and proliferation [125] of VSMCs, thus demonstrating that the ET-1 pathway is also implicated in remodeling processes leading to vascular occlusion.

VEGF is responsible for neoangiogenesis, which increases the recruitment of other immune cells in the arterial wall [127]. Illustrating this neoangiogenesis process, *vasa vasorum*, which are physiologically restricted to the adventitia, are observed in the media and intima of arteries affected by GCA and correlate with internal elastic lamina digestion and infiltration by giant cells. IL-33, which is an alarmin belonging to the IL-1 family, is overexpressed in GCA arteries and also involved in angiogenesis [128].

## 5. Role of B Cells

In GCA, B cells are much rarer than T cells in the arterial wall. They are localized in the adventitia and the media next to CD3^+^ cells in structures defined as artery tertiary lymphoid organs (ATLOs) [129]. Studies showed an association with ectopic expression of CXCL13 and B-cell activation factor (BAFF), which increases after in vitro stimulation of temporal arteries with IL-6 [130]. Furthermore, a recent study showed that chemokines CXCL9 and CXCL13 are increased in the circulation of untreated GCA and PMR patients and that peripheral CXCR3^+^ and CXCR5^+^ switched memory B cells are significantly reduced in GCA and PMR compared to healthy controls and inversely correlate with the serum levels of their complementary chemokines CXCL9 and CXCL13. Suggesting the implication of these chemokines in the recruitment of B cells in the arterial wall, CXCR3^+^ and CXCR5^+^ B cells were observed in areas with high expression levels of CXCL9 and CXCL13 [131].

Nevertheless, the humoral immune response does not appear to play as important a role in GCA as in Takayasu arteritis, which is the other primary large-vessel vasculitis. Indeed, a recent study found an increased TFH signature in both circulating and aortic CD4 T cells in Takayasu arteritis but not in healthy controls or GCA patients. This highlights that the cooperation between T and B cells is probably more critical in Takayasu arteritis than in GCA [132]. Another study that analyzed the involvement of mammalian target of rapamycin (mTOR) in large-vessel vasculitis suggested that humoral immunity plays an important role in Takayasu arteritis but not GCA [133]. Indeed, this study demonstrated that mTOR complexes were activated in endothelial cells from Takayasu patients but not GCA patients, and that there were higher levels of antibodies binding to endothelial cells in Takayasu patients compared to GCA. Purified antibodies from Takayasu patients caused mTOR activation and significant endothelial cell proliferation, which was not the case with antibodies from GCA patients [133].

## 6. Mechanisms Involved in Maintaining Inflammation

As discussed above, the mechanisms involved in inflammation and vascular remodeling are becoming better understood and are promising therapeutic targets. In addition, some mechanisms allow inflammation to become chronic, which could also represent interesting therapeutic targets to replace glucocorticoids.

### 6.1. Defect in Immune Checkpoints Inhibitors

Programmed death-1 (PD-1) is a surface protein expressed by activated T cells, and its ligation to PD-L1 or PD-L2, which are expressed by antigen-presenting cells, induces T-cell apoptosis, T-cell anergy and the production of IL-10 by T cells or their polarization into Treg. By contrast, PD-1^−/−^ mice, in which the PD-1/PD-L1 pathway was lacking, displayed elevated Th1 and Th17 levels [134]. In GCA, a defect in the immunoprotective PD-1/PD-L1 immune checkpoint has recently been reported. This deficit relates to the lower expression of PD-L1 by vascular DCs, which sustain IL-17-, IL-21- and IFN-γ-producing PD-1^+^ T cells and the emergence of typical lesions of GCA, such as intimal hyperplasia and neoangiogenesis [135]. Along this line, it was shown that the proportions of PD-1^+^ and VISTA^+^ (a negative immune checkpoint V-domain immunoglobulin-containing suppressor of T cell activation) T cells were decreased in the blood of GCA patients because they were recruited in GCA lesions. However, contrasting with PD-1/PDL-1, VISTA-Ig engagement failed to suppress Th1, Th17 and TFH lineage development in GCA [136].

The implication of immune checkpoints in the GCA pathogenesis was also highlighted by the recently demonstrated efficacy of abatacept for the treatment of GCA [137]. Abatacept is a fusion protein composed of cytotoxic T-lymphocyte-associated protein 4 (CTLA4) and the fragment crystallizable region of a human IgG_1_. Due to the competitive binding of CTLA4 to CD80/CD86, abatacept dampens T-cell activation by impeding the interaction between CD28 and CD80/86. In contrast, the occurrence of PMR/GCA has recently been reported in patients treated with antagonists of CTLA-4 or PD-1 used in metastatic melanoma, ipilimumab and nivolumab [138,139].

### 6.2. Defect in Regulating T Cells (Tregs)

Tregs are major actors in immune tolerance and protect patients from inflammation and autoimmunity [140]. Our team reported a quantitative defect in circulating Tregs (CD4^+^CD25^high^FoxP3^+^) together with their absence from arterial lesions of GCA [36]. The Treg population is heterogeneous and they are best defined by their functional activity [141]. Confirming the role of a defective T regulatory response in GCA, two studies recently demonstrated that GCA patients had a Treg compartment enriched in IL-17 secreting a Treg (Th17-like Treg) with an impaired suppressive capacity, which was mainly related to the expression of a hypofunctional isoform of FoxP3 lacking exon 2 (FoxP3Δ2), which is required for the antagonization of RORγt and RORα, the main transcriptional factors controlling the production of IL-17 [101,142]. The blockade of the IL-6 pathway with tocilizumab normalized the population of FoxP3Δ2 Treg, thus suggesting that IL-6 is involved in this defect [101]. In addition, we demonstrated that, unlike healthy controls, Tregs from GCA patients increased the polarization of T cells toward Th17 cells, which was corrected after in vitro treatment with tocilizumab [101]. Another study compared the transcriptomic signatures of Tregs from healthy controls and GCA patients in the active phase or in remission. The results, which were confirmed by functional experiments, showed that the calcium influx and glycolysis were severely impaired in Tregs from GCA patients, that these abnormalities correlated with Treg dysfunction in GCA and that tocilizumab could not correct this defect in the calcium influx [143].

As for CD4^+^ T cells, a defect in CD8^+^ regulatory T cells was also reported in GCA [46]. CD8 Tregs, which are defined by a CD8^+^CCR7^+^FoxP3^+^ phenotype, control the proliferation and activation of CD4^+^ T cells through the production of exosomes containing NADPH oxidase 2 (NOX2). These exosomes are captured by neighboring CD4^+^ T cells, in which NOX2 disrupts activation, survival and proliferation pathways [46]. Interestingly, aging is associated with a progressive loss of the expression of NOX2 by CD8 Treg; this decrease is even more important in GCA and not corrected by glucocorticoids [46].

### 6.3. Implication of Other Subsets of T Cells

Our team recently investigated the implication of mucosal-associated invariant T cells (MAIT) in GCA. MAIT cells are innate-like lymphocytes characterized by the expression of a semi-invariant T-cell receptor (TCR) composed of a constant α chain (TCRVα7.2-Jα33) and a β chain (among a limited number of variants, often Vβ2 and Vβ13). Interaction between the two is restricted to major histocompatibility complex (MHC)-related protein 1 (MR1). MAITs display an immediate effector function on stimulation, and they have a high clonal volume [144]. It has also been established that MAIT cells are activated during viral infections by a TCR-independent pathway [145,146,147]. This particular means of activating MAIT cells involves IL-12 and IL-18 [148], two cytokines that are highly expressed in GCA lesions [84,148], which led us to hypothesize that MAIT cells could be involved in the GCA pathogenesis, as is the case in ANCA-associated vasculitis [149]. MAIT cells were found in the arterial wall of GCA temporal arteries but were absent from healthy arteries. The MAIT frequency was similar in the blood of patients and controls, but the level of expression of IFN-γ was increased in MAIT cells from GCA patients, and when they were stimulated with IL-12 and IL-18, MAIT from GCA patients produced very high levels of IFN-γ and displayed stronger proliferation compared to MAIT from controls. Thus, MAIT could be involved in the maintenance of the inflammatory response during GCA through its ability to be activated by IL-12 and IL-18 without TCR signaling, especially since these two cytokines are present in GCA lesions [84,150].

Another study also suggested a role of tissue-resident memory T cells (TRMs) in chronic inflammation. Specialized to remain resident in the tissue microenvironment, TRMs are thought to provide rapid and effective immune responses when they re-encounter antigens, and they are considered pathogenic in chronic inflammatory diseases. They have a pro-inflammatory effector function and secrete pro-inflammatory cytokines such as IFN-γ, IL-17, IL-9 and TNF-α. In GCA arteries, there is a population of CD4^+^CD103^+^ memory T cells that are barely detectable in the peripheral blood. This TRM population is sustained by IL-7, IL-9 and IL-15, which are present in the tissue microenvironment of GCA lesions. TRMs require JAK3/1-dependent signaling through γc chain-containing cytokine receptors to survive. Along this line, researchers demonstrated that the inhibition of JAK1/3 activity with tofacitinib minimized the in situ proliferation of CD4^+^CD103^+^ TRM. The persistence of vascular wall inflammation in GCA arteries may, therefore, also depend on a small, highly specialized population of CD4+CD103+ TRMs characterized by their ability to survive in the tissue microenvironment [100].

### 6.4. Role of VSMCs

VSMCs are major components of the vessel wall. They are characterized by contractile functions, a synthetic function that produces components of the extracellular matrix and they are also involvement in tissue repair through their ability to migrate and proliferate [94]. In GCA, VSMCs are not only targets but also actors in the inflammatory response. In addition to their ability to migrate and proliferate, which are fundamental to vascular remodeling, when exposed to IFN-γ, these cells produce chemokines that enhance the recruitment of new T cells and monocytes [107]. This sets up an amplification loop that will contribute to the development of vascular inflammation. In addition to this function, we have data in the process of publication showing that myofibroblasts can interact with T cells and maintain their Th1 polarization.

### 6.5. Implication of the NOTCH Pathway

The interplay between T cells and resident cells of the arterial wall also involves the NOTCH pathway, a highly conserved cell-signaling system that is important for cell-cell communication, gene regulation mechanisms controlling cell differentiation processes during embryonic and adult life, and the development of blood vessels. NOTCH receptors and their ligands (Jagged and Delta) are transmembranous proteins. The ligation of NOTCH to Jagged or Delta ligands triggers proteolytic events leading to the translocation of the intracellular domain of NOTCH into the nucleus, where it interacts with transcription factors regulating the destiny of cells [151,152]. Several NOTCH receptors and their ligands are expressed in healthy arteries where they regulate differentiation, along with plasticity of the VSMC phenotype, and facilitate the cross-talk between VSMCs and endothelial cells [153]. VSMCs and endothelial cells express NOTCH receptors and their ligands, as well as CD4^+^ T cells that express Notch 1 and Jagged 2 [154]. In GCA, abundant VEGF in the blood upregulates the expression of Jagged 1 by adventitial microvascular endothelial cells, allowing effector T-cell induction via the Notch-mTORC1 pathway. CD4 T cells in GCA patients differentiate into Th1 and Th17 effector cells through the NOTCH pathway, and in an in vivo model of large-vessel vasculitis, exogenous VEGF functioned as an effective amplifier to recruit and activate vasculitogenic T cells [155]. Furthermore, Notch 1 was 20 times higher in T cells from GCA patients than those from healthy controls, which allowed them to interact with DCs, macrophages, VSMCs and endothelial cells expressing Notch 1 ligands [156]. The blockade of the NOTCH pathway with γ-secretase treatment in a mouse model of GCA strongly depleted Th1 and Th17 cells from the vascular infiltrates, thus showing the implication of this pathway in the GCA pathogenesis [156]. Furthermore, a recent paper showed that in GCA, the molecular defect of malfunctioning CD8^+^Treg cells lies in aberrant Notch 4 signaling that deviates endosomal trafficking and minimizes exosome production. By transcriptionally controlling the profile of RAB GTPases, Notch 4 signaling restricts the vesicular secretion of the enzyme NADPH oxidase 2 (NOX2) [157].

### 6.6. Role of Granulocyte-Macrophage Colony-Stimulating Factor (GM-CSF)

Emerging as a key cytokine in inflammation, granulocyte-macrophage colony-stimulating factor (GM-CSF) may play a role in promoting inflammation in GCA. GM-CSF contributes to the pathophysiology of GCA by regulating inflammatory macrophages, DCs and Th1 and Th17 cells and is involved in angiogenesis and vascular remodeling [112,158,159,160,161]. A recent study demonstrated that GM-CSF and GM-CSF receptor α (GM-CSFRα) transcripts and proteins were highly expressed in GCA vascular lesions and that macrophages and pericytes were the main sources of GM-CSF in GCA lesions [160]. Moreover, signaling pathways activated by GM-CSF (JAK2, STAT5A) were shown to be activated in GCA lesions [160]. Using a model of an ex vivo culture of temporal arteries treated with GM-CSF or anti-GM-CSF (mavrilimumab), researchers showed that GM-CSF increased the activation of macrophages (expression of IL-1β, IL-6, TNF-α, CD83 and HLA-DR), Th1 cell polarization, angiogenesis and tissue injury (MMP9/TIMP1 ratio) [160]. These results led a phase-2, randomized, placebo-controlled therapeutic trial to be conducted, the results of which strongly suggest the efficacy of mavrilimumab, a fully human IgG4 monoclonal antibody targeting GM-CSF, in the treatment of GCA [162].

### 6.7. Role of Interleukine-6 (IL-6)

IL-6 is produced by many cells, especially monocytes and macrophages, and has pleiotropic effects. IL-6 signaling depends on gp130, a transmembranous glycoprotein that triggers the phosphorylation of STAT3. Gp130 is activated through its ligation to a complex composed of IL-6 and its receptor, either membranous (mIL-6R) or soluble (sIL-6R). While the expression of gp130 is ubiquitous, the expression of mIL-6R is restricted to hepatocytes, monocytes, macrophages, a few B and T cells, megakaryocytes and endothelial cells [163]. Classical signaling of IL-6 involves mIL-6R, whereas trans-signaling involves sIL-6R, the latter being of particular importance among the pro-inflammatory functions of IL-6 [163].

In GCA, the concentration of serum IL-6 is very high and correlates positively with disease activity [7,36]. The major role of IL-6 in the GCA pathogenesis is demonstrated by the dramatic efficacy of tocilizumab for the treatment of GCA [78,164]. Furthermore, the Th17/Treg imbalance observed in GCA [36] is controlled by IL-6, which increases Th17 polarization and decreases Treg differentiation [102]. IL-6 is also involved in the recruitment of leukocytes in the arterial wall. When they are exposed to IL-6, endothelial cells that express mIL-6R and gp130 express adhesion molecules such as VCAM-1 and ICAM-1, thus leading to the recruitment of leukocytes expressing VLA-4 and LFA-1 by increasing their attachment and transendothelial migration [165]. Furthermore, the intercellular IL-23p19 peptide, produced in endothelial cells in GCA and promoted by pro-inflammatory factors (LPS, TNF-α and IFN-γ), stimulates the gp130-dependent activation of STAT3 by its association with the cytokine receptor subunit gp130, thus leading to an increase in intercellular adhesion molecules at the cell surface. Therefore, IL-6 triggers the amplification of the inflammatory processes involved in the GCA pathogenesis [165]. In contrast, IL-6 does not appear to be involved in vascular remodeling. A recent study demonstrated that despite an increase in VEGF after IL-6 treatment of temporal artery explants from GCA patients, there was no increase in myofibroblast proliferation or migration after IL-6 treatment [166].

## 7. Conclusions

Knowledge of the mechanisms involved in the pathogenesis of GCA has improved considerably in recent years, leading to the identification of new therapeutic targets to improve patient treatment and reduce the use of glucocorticoids. The most recent work also showed the involvement of processes leading to the amplification and/or maintenance of inflammation and vascular remodeling, such as regulatory T response defects (regulated by IL-6), GM-CSF and the roles of macrophages and NOTCH, along with probably a major role of arterial wall resident cells. These mechanisms identify targets for the development of new therapeutics such as tocilizumab [77], and more recently, mavrilimumab [162], secukinumab [167], JAK inhibitors [168], abatacept [137] and ustekinumab [169,170,171] (Figure 3) [172].

## Figures and Tables

**Figure 1 jcm-11-02905-f001:**
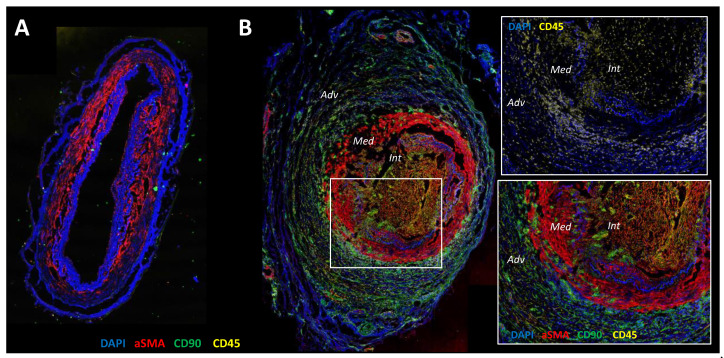
Confocal microscopy analysis of healthy temporal artery (**A**) and GCA temporal artery (**B**) with staining of α-SMA (red), CD90 (green), CD45 (yellow) and nuclei (DAPI (4′,6-Diamidine-2′-phenylindole dihydrochloride), blue). (**A**): The arterial wall of the healthy artery is well preserved. DAPI underlines collagenic structures such as internal and external elastic lamina. Vascular smooth muscle cells (α-SMA^+^) are restricted in the media and the intima is very thin. (**B**): The Giant Cell Arteritis (GCA) artery is characterized by inflammation and severe vascular remodeling. The zoomed-in square region shows CD45 staining of an infiltration of the arterial wall by mononuclear cells. The adventitia is rich in CD90^+^ cells that are fibroblasts, the internal elastic lamina and media are digested and there α-SMA^+^CD90^+^ cells in the intima, which fits with myofibroblasts that have migrated and proliferated from the media to the intima, resulting in severe intimal hyperplasia and leading to the stenosis of the vascular lumen.

**Figure 2 jcm-11-02905-f002:**
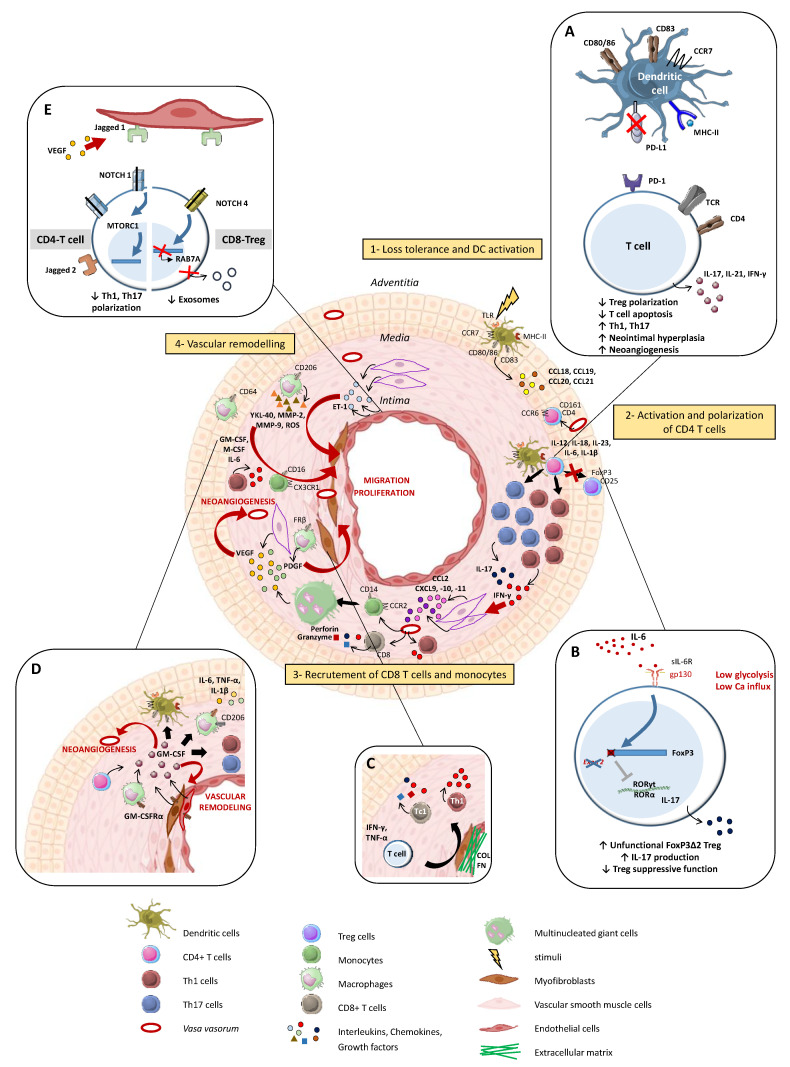
Summarized pathogenesis of Giant Cell Arterits (GCA). Step 1: an undefined danger signal activates vascular dendritic cells (DC) that then acquire a mature phenotype (CD83^+^CD80/86^+^CCR7^+^MHC-II^high^) and produce chemokines (CCL18, CCL19, CCL20 and CCL21), leading to the recruitment of CCR6^+^CD161^+^CD4^+^ T cells. Step 2: CD4^+^ T cells are activated by DCs and polarize into Th1 and Th17 cells through the effect of IL-12, IL-23, IL-6 and IL-1β, which are produced by activated DC. Th1 and Th17 lymphocytes release IFN-γ and IL-17, respectively. Step 3: IFN-γ induces the activation of vascular smooth muscle cells (VSMC) in the media and enables them to produce chemokines (CCL2, CXCL9, CXCL10, CXCL11), which trigger the recruitment of additional T cells (CD4^+^ and CD8^+^) and monocytes. Monocytes differentiate into macrophages and merge into multinucleated giant cells, the hallmark of GCA. Step 4: vascular remodeling is characterized by the destruction of the internal elastic lamina and the proliferation and migration of VSMC into the intima. Macrophages play a key role in this process through the release of several factors such as Platelet-Derived Growth Factor (PDGF), reactive oxygen species (ROS), Matrix metalloproteinase-9 (MMP-9), IL-6, IL-1β, Granulocyte-Macrophage Colony-Stimulating Factor (GM-CSF) and TNF-α, which contribute to tissue damage and intimal hyperplasia. Likewise, VSMCs and endothelial cells release ET-1, which stimulates VSMC migration and proliferation and thus induces intimal hyperplasia. Moreover, macrophages and VSMCs also produce vascular endothelial growth factor (VEGF), which is responsible for neoangiogenesis and promotes a local inflammatory response. A cellular transition from VSMC to a myofibroblast phenotype is observed, and the accumulation of these cells in the neo-intima leads to vascular occlusion, which is responsible for ischemic complications of GCA. (**A**–**E**) Mechanisms involved in the maintenance of vascular inflammation. (**A**): Programmed death-ligand 1 (PDL-1) defect on antigen-presenting cell surface leads to the persistent activity of T cells and contributes to hyperplasia and neoangiogenesis. (**B**): IL-6, which is a pro-inflammatory cytokine implicated in the GCA pathogenesis, impairs Tregs’ function, decreases their frequency and promotes the shift to a Treg deficiency in exon 2 of FoxP3 that is prone to producing IL-17. (**C**): In addition to their role in the recruitment of monocytes and vascular remodeling, VSMC differentiate into myofibroblasts, which also participate in the maintenance of Th1 and Tc1 polarizations. Moreover, myofibroblasts have an important capacity to produce extracellular matrix proteins (COL: collagen, FN: fibronectin) that contribute to the rigidification of the vascular wall. (**D**): GM-CSF is produced by CD4 T cells, macrophages, VSMC and endothelial cells. GM-CSF-Receptor-α is highly expressed in GCA lesions, and an autocrine amplification loop takes place. GM-CSF is involved in cell differentiation, vascular inflammation and vascular remodeling. (**E**): T cells and resident cells of the arterial wall, like endothelial cells, communicate via the NOTCH pathway. The ligation of Notch 1 to Jagged 1 decreases the polarization Th1/Th17. Otherwise, in CD8^+^Treg cells, aberrant Notch 4 signaling drives the suppression of RAB7A involved in the exosomal release of NOX2.

**Figure 3 jcm-11-02905-f003:**
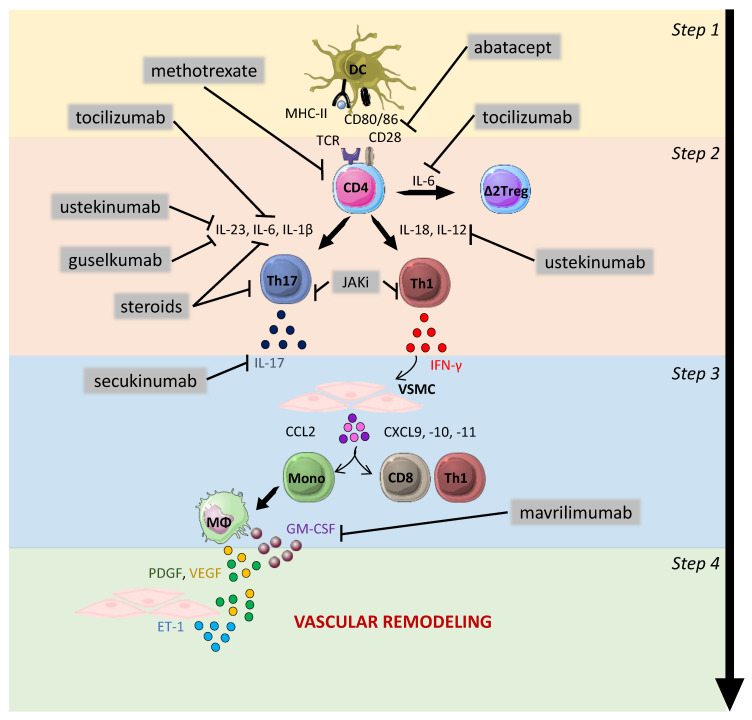
Therapeutic strategies in GCA. Steps 1–4 show the pathogenesis of the GCA physiopathology: 1—activation of dendritic cells; 2—activation, proliferation and polarization of T cells toward Th1 and Th17 cells; 3—effect of interferon-gamma (IFN-γ) on vascular smooth muscle cells (VSMCs) leading to the recruitment of additional CD4 T cells together with CD8 T cells and monocytes; 4—production of mediators implicated in vascular remodeling: neoangiogenesis (VEGF), migration, proliferation and differentiation of VSMCs into myofibroblasts, leading to hyperplastic neointima. The main drugs approved or under evaluation for the treatment of GCA are shown in the figure with their main therapeutic targets. Steroids inhibit T-cell activation, proliferation and polarization into Th17 cells. In addition, they trigger a decrease in the level of serum IL-6. Abatacept blocks T-cell activation through its ability to prevent interaction between CD28 and CD80/86. Methotrexate inhibits the proliferation of T cells and their ability to produce cytokines. Ustekinumab targets the p40 subunit, which is shared by IL-12 and IL-23. Guselkumab targets the p19 subunit of IL-23. Tocilizumab targets the receptors of IL-6. Mavrilimumab targets GM-CSF and should, therefore, impact vascular remodeling. Janus kinase inhibitors (JAKi) block the signaling pathways of several cytokines such as IL-6, IL-12 and IFN-γ and can thus theoretically inhibit the Th1 and Th17 pathways, to inhibit vascular inflammation and remodeling. DC: dendritic cells; VSMC: vascular smooth muscle cells; Mono: monocytes; MΦ: macrophages; PDGF: platelet-derived growth factor; VEGF: vascular endothelial growth factor; ET-1: endothelin-1; GM-CSF: granulocyte-macrophage colony-stimulating factor.

## Data Availability

Not applicable.

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
