# Peer review of "New Insights into the Pathogenesis of Giant Cell Arteritis: Mechanisms Involved in Maintaining Vascular Inflammation"

_jcm, 2022, doi:10.3390/jcm11102905_

Round 1
Reviewer 1 Report
Thank you for giving me the opportunity to review an in-depth and broad narrative review in GCA pathogenesis. It covers all stages of GCA pathogenesis in great detail, using up-to-date references. The figures are well-illustrated and informative. Only minor comments:
- Lines 38-39, “that is largely correlated with serum interleukin-6 (IL-6) elevation”: please provide reference
- Lines 104-105: Use the term “T cells” instead of “T lymphocytes”
- Line 107: Please correct the content of parenthesis “(CD3, CD3, CD3, CD3)”
- Lines 120, 167, 580: In line 120, you write “DC”, but “DC” is defined only later in lines 167 AND line 580. Please correct it by defining “DC” only in line 120. In general, recheck all abbreviations
- Lines 155-156, “VZV vasculitis can also affect the temporal 155 arteries and cause GCA-like symptoms”: The authors would better add a reference and my suggestion is
“Cranial giant cell arteritis mimickers: A masquerade to unveil. Evangelatos G, Grivas A, Pappa M, Kouna K, Iliopoulos A, Fragoulis GE. Autoimmun Rev. 2022 Mar 24;21(5):103083. doi: 10.1016/j.autrev.2022.103083. PMID: 35341973”
Author Response
We thank reviewer 1 for his careful proofreading. All comments have been taken into consideration and corrected.

Reviewer 2 Report
Overall, the paper is well written and compiles the findings on the pathogenesis of GCA.
But, due to its many details, the paper is generally difficult to read and understandable only for specialists. Here, more figures could contribute to a much better understanding of the text.
The authors should also highlight the implications for new therapeutic approaches (e.g. phase II studies with mavrilimumab or secukinumab, among others).
There are different histological types of GCA, not only a panarteriitis or transmural vasculitis, but also a vasculitis of vasa vasorum, adventitial small-vessel vasculitis (Cavazza et al. American J. Surgical Pathology 2014; Cicca et al. Rheumatology 2015). These different types differ also in their cytokine profile. This should be addressed in the paper.
Author Response
Overall, the paper is well written and compiles the findings on the pathogenesis of GCA.
But, due to its many details, the paper is generally difficult to read and understandable only for specialists. Here, more figures could contribute to a much better understanding of the text. The authors should also highlight the implications for new therapeutic approaches (e.g. phase II studies with mavrilimumab or secukinumab, among others).
We thank reviewer 2 for his valuable comments. To enhance the understanding of the article, we have added an additional figure (Figure 3) showing in a simplified manner the pathophysiology of GCA and the main therapeutic targets. We have chosen not to add more details in the text because the treatment of GCA is discussed in another review in the same special issue for which this article is intended [1].
There are different histological types of GCA, not only a panarteriitis or transmural vasculitis, but also a vasculitis of vasa vasorum, adventitial small-vessel vasculitis (Cavazza et al. American J. Surgical Pathology 2014; Cicca et al. Rheumatology 2015). These different types differ also in their cytokine profile. This should be addressed in the paper.
We thank reviewer 2 for this interesting comment. A paragraph mentioning these data have been added in the manuscript (see below).
“Interestingly, the predominant cytokine profile could have a role in the emergence of the various histological patterns of GCA [2,3]. Indeed, it was reported that IL-17 overexpression was observed mainly in arteries with transmural inflammation and vasa vasorum vasculitis whereas IL-9 and Th9 polarization was predominant in case of transmural inflammation and small-vessel vasculitis [4].”
REFERENCES
- Regent, A.; Mouthon, L. Treatment of Giant Cell Arteritis (GCA). Journal of clinical medicine 2022, 11, doi:10.3390/jcm11071799.
- Delaval, L.; Samson, M.; Schein, F.; Agard, C.; Trefond, L.; Deroux, A.; Dupuy, H.; Garrouste, C.; Godmer, P.; Landron, C.; et al. Temporal Arteritis Revealing Antineutrophil Cytoplasmic Antibody-Associated Vasculitides: A Case-Control Study. Arthritis Rheumatol 2020, doi:10.1002/art.41527.
- Cavazza, A.; Muratore, F.; Boiardi, L.; Restuccia, G.; Pipitone, N.; Pazzola, G.; Tagliavini, E.; Ragazzi, M.; Rossi, G.; Salvarani, C. Inflamed temporal artery: histologic findings in 354 biopsies, with clinical correlations. Am J Surg Pathol 2014 Oct, 38, 1360-1370.
- Ciccia, F.; Rizzo, A.; Guggino, G.; Cavazza, A.; Alessandro, R.; Maugeri, R.; Cannizzaro, A.; Boiardi, L.; Iacopino, D.G.; Salvarani, C.; et al. Difference in the expression of IL-9 and IL-17 correlates with different histological pattern of vascular wall injury in giant cell arteritis. Rheumatology (Oxford) 2015, 54, 1596-1604, doi:10.1093/rheumatology/kev102.

Round 2
Reviewer 2 Report
there are no comments
Author Response
N/A